# Exploring Structure-Property Relationships in a Bio-Inspired Family of Bipodal and Electronically-Coupled *Bis*triphenylamine Dyes for Dye-Sensitized Solar Cell Applications

**DOI:** 10.3390/molecules25092260

**Published:** 2020-05-11

**Authors:** Tamara Al-Faouri, Francis L. Buguis, Saba Azizi Soldouz, Olga V. Sarycheva, Burhan A. Hussein, Reeda Mahmood, Bryan D. Koivisto

**Affiliations:** Department of Chemistry and Biology, Ryerson University, 350 Victoria St, Toronto, ON M5B 2K3, Canada; tamara.alfaouri@ryerson.ca (T.A.-F.); fbuguis@uwo.ca (F.L.B.); saba.azizisoldouz@ryerson.ca (S.A.S.); sarycheva.o@gmail.com (O.V.S.); burhan.a.hussein@durham.ac.uk (B.A.H.); r4mahmoo@ryerson.ca (R.M.)

**Keywords:** structure–property relationships, absorption and fluorescence, cyclic voltammetry, DSSC device fabrication, synthesis, π-spacer effects, triphenylamine substituent effects

## Abstract

A bio-inspired family of organic dyes with bichromic-bipodal architectures were synthesized and tested in dye-sensitized solar cells (DSSC). These dyes are comprised of a D-π-D-A motif with two triphenylamine (TPA) units acting as donors (D) and two cyanoacetic acid acceptors (A) capable of binding to a titania semiconductor. The role of the thiophene π-spacer bridging the two TPA units was examined and the distal TPA (relative to TiO_2_) was modified with various substituents (-H, -OMe, -SMe, -OHex, -3-thienyl) and contrasted against benchmark **L1**. It was found that the two TPA donor units could be tuned independently, where π-spacers can tune the proximal TPA and R-substituents can tune the distal TPA. The highest performing DSSCs were those with -SMe, 3-thienyl, and -H substituents, and those with one spacer or no spacers. The donating abilities of R-substituents was important, but their interactions with the electrolyte was more significant in producing high performing DSSCs. The introduction of one π-spacer provided favourable electronic communication within the dye, but more than one was not advantageous.

## 1. Introduction

While much of the research in next-generation photovoltaics has shifted towards perovskite solar cells, owing to their high performance [1,2,3,4,5], the dye-sensitized solar cell (DSSC) remains a promising technology for light harvesting [6,7,8]. The DSSC continues to demonstrate a flexible device architecture [9,10] that is useful for untraditional light environments (diffuse light, optically transparent and niche coloured applications, etc.) [11,12,13,14,15,16]. Furthermore, the use of organic dyes containing a donor, π-spacer, acceptor (D-π-A) motif have been shown to facilitate the intramolecular charge transfer necessary for charge extraction and performance within the DSSC [17,18,19,20]. 

Perhaps one of the most common donors used in organic DSSC dyes is the triphenylamine (TPA) moiety due to its electron rich nature, redox stability and non-planarity, which reduces intermolecular dye aggregation [21,22,23,24,25,26,27]. The structural diversity possible with TPA has afforded a number of organic dye topologies [28,29,30,31,32,33,34,35,36] and each modification can lead to a significant difference in optical and electronic properties. In a previous contribution, our group presented a family of D-π-D-π-A bichromic/bipodal dyes (two TPAs and two cyanoacetic acid anchors) and we investigated the placement (and importance) of a thiophene π-spacer [30]. It was hypothesized that the thiophene placement could be critical for ground state electronic coupling between the TPA donors. In addition, having the donors and acceptors bridged by a thiophene unit could increase the panchromatic absorption, redox and dye stability, and the overall electronic communication between interfaces (TiO_2_/dye and dye/electrolyte) [37,38,39,40,41,42,43,44]. It was found that the location of the thiophene π-spacer between the donor and acceptor anchors did not enhance optical properties nor the device performance, while significantly increasing synthetic effort. However, there was an improvement in performance when a single thiophene π-spacer bridged the TPA donors. 

Building on this initial report, we now present a more extensive family of bio-inspired dyes (Figure 1). While the comparison between biology and the DSSC has been made before [45], our dye design attempts to mimic the dimeric structure of photosystem II (PSII). In PSII, two quinoidal routes exist (akin to the bipodal acceptor motif) that can extract charge from the reaction centre. The reaction centre (P680) is comprised of two cofacial porphyrin rings that create an electronically-coupled redox-active environment (akin to two TPA donors). In this study, we now delve deeper into the structure–property relationships involving the nature of the π-spacer between the TPA units and the electronic tuning of the distal (from anchors) TPA. In our previous study, it was unclear if the enhancement in dye performance was due to redox interplay of the two donors, or increased conjugation and hydrodynamic volume of the dye. To this end, we present a modified dye topology with a D-π-D-A motif that has greater control on the redox potential of the distal TPA unit, while contrasting the electronic separation of the donors.

## 2. Dye Design and Synthesis

Our synthesized family of bichromic-bipodal D-π-D-A dyes are presented in Figure 2 and organized into three groups. The full synthetic procedures and characterization of these dyes and 16 novel precursors inspired from previously reported building blocks [46,47,48,49,50] can be found in the Appendix A. Dyes in Group 1 do not have any thiophene π-spacers, whereas, dyes in Group 2 contain one thiophene π-spacer between the two TPA units, and dyes in Group 3 contain two thiophene π-spacers between the two TPA units. In addition, each group contains dyes with no substituents -H (a), or -OMe (b), -OHex (c), and -SMe (d) substituents on the distal TPA unit. Dye **4**, represents a different distal TPA modification (3-thienyl) and is an extension of the Group 1 family and previous work [51]. **L1** is a previously reported dye [52] that we use as a benchmark. 

In our previous report [30], distal TPA modification was restricted to -OMe, owing to its ease of synthesis and ability to red-shift the absorption profile by raising the HOMO energy of the donor. In addition, when moving the thiophene spacer to create different motifs (e.g., D-π-D-π-A) only subtle redox tuning is needed to significantly affect dye performance in these *bis*-TPA dyes. Therefore, in this study we included the weaker donor -SMe and no substituents at all (-H). Thiomethylether attachments have also been shown to improve dye regeneration by the electrolyte [53]. While hexylether-substituents do not alter the redox properties differently than -OMe, they are frequently used to improve solubility and prevent dyes from aggregating on the titania surface [54,55]. However, in this study we were keen to see if that would be relevant with this rigid bipodal structure. Having two attachments to the titania surface will restrict the degrees of freedom of the dye and likely prevent aggregation and close proximity between dyes on the surface. Finally, while it may appear as an outlier, molecule **4** uses a 3-thienyl substituent to increase the optical cross section of the distal donor, while only modestly altering the redox properties and builds on our previous work of creating polymerizable DSSC dyes [51].

## 3. Results and Discussion

### 3.1. Optical Behaviour

All physicochemical properties of the dyes discussed herein were collected in dichloromethane solutions and are collated in Table 1. In addition, the complete compendium of UV-Vis absorption (Appendix A) and fluorescence spectra (Appendix A) are included in the Appendix A. All the dyes (with the exception of **4**) exhibit weak or negligible fluorescence behavior in solution, consistent with previous observations [31,51]. In addition, time-dependent density functional theory (TD-DFT) calculations were performed to better appreciate the dominant optical transitions. Ground-state geometries were optimized with the B3LYP hybrid functional. All optimizations were calculated without any symmetry constraints using the 6-31G(d,p) basis set with the Gaussian16 software package [56].

Upon converting the aldehyde precursors into the cyanoacetic acid dye derivatives **1**–**4**, there is a significant broadening and bathochromic shift in the UV-Vis absorption spectra for all dyes. This phenomenon seemed to be most pronounced in the Group 1 family of dyes (~100 nm for Group 1, and ~50 nm for Group 2 and 3, respectively). This more pronounced effect in Group 1 could suggest that there is a more significant difference between the ground and excited state structure in this family. In addition, the Group 1 dyes possess multiple maxima in their spectra when compared to the broader profiles of Group 2 and 3 dyes. The presence of thiophene π-spacers, which generally increases the π-conjugation, causing a more significant bathochromic shift does not seem to be observed in this family, as all the low energy absorption edges tend to become negligible around 525 nm. This observation is consistent with the previous study on similar dyes [30]. Furthermore all -SMe derivatives also possess a high energy maxima at ~325 nm. Table 1 summarizes the maximum absorbance wavelengths (λ_max_) and the molar extinction coefficients (ε) for each dye studied.

In order to better appreciate the optical behaviour imparted by our structural variations, TD-DFT calculations were performed and their insight is included in Figure 3, Figure 4 and Figure 5. In each of the dyes studied, HOMO-LUMO optical transitions are not observed because of insufficient wavefunction overlap between these orbitals and a resulting weak transition oscillator strength. Considering the substitution on the distal TPA (Figure 3), there is a minimal effect on the visible portion of the absorption profile. Although there were more significant differences at shorter wavelengths (~325 nm), bathochromic shifts at longer wavelengths were not observed by the introduction of different R substituents. While perhaps a little disappointing, this optical parity is actually quite significant, because it allows us to focus on the electronic effects these substituents impart on the dye (and device performance). A subtle change was observed in the visible region (400–500 nm) as the two maxima in **1b** appear less resolved. This is rationalized due to the decreased oscillator strength for the HOMO to LUMO + 2 transition and an increase in the HOMO − 1 to LUMO + 1 contribution. It is worth noting that in all cases the electron density in the HOMO level was at the distal TPA, and the LUMO and LUMO + 1 on the proximal TPA. In the LUMO + 2 the electron density was shared between the two TPAs. The electron density in the HOMO − 1 was also concentrated on the proximal TPA with some delocalization throughout the molecule.

The presence of thiophene π-spacers had a more pronounced effect on the spectra, resulting in them becoming less resolved and absorption maxima becoming hypsochromically shifted (Figure 4). In all cases, the electron density of the HOMO orbital was localized on the distal TPA in the LUMO localized on the proximal TPA (and anchors). In the LUMO+2 electron density was localized between the two TPA units. While the long wavelength absorption edge was the same for zero and one π-spacer, the two-spacer derivatives had blue-shifted and less resolved spectra, without a significant increase in the extinction coefficient. While this may be a deleterious effect in terms of device efficiency, the decrease in resolution is due to the increased π-conjugation from the thiophene π-spacers. As the number of thiophene spacers increase, so too does the contribution from the HOMO to LUMO + 1 and HOMO − 1 to LUMO + 1 transitions. This manifests as a less resolved peak. This would suggest enhanced electronic communication between the redox-active (TPA) units, and a decrease in their independent optical behaviour. As the π-conjugation increases, the absorption profile normally undergoes a bathochromic shift in D-π-A systems [18]. However, our unique systems are generally described as D-π-D-A, and in this situation, coalescence of absorption maxima is observed as we extend conjugation between the two donors. 

The effect of having a 3-thienyl substituent on the distal TPA was also examined optically by contrasting dye **4** with **1a** (Figure 5). In both cases the LUMO electron density is localized on the proximal TPA (and anchor) and HOMO electron density localized on the distal donor; however, HOMO delocalization on the 3-thienyl substituent does occur in **4**. The absorption profile of **4** possessed a lower ε compared to **1a** at longer wavelengths, but an increase at higher energy. TD-DFT calculations predict that this hyperchromic behaviour at short wavelengths (330 nm) is due to a transition from HOMO to LUMO + 4 (not shown, but heavily localized on the thiophenes) but is unlikely to have a significant effect on device performance (owing to the filters used in our solar simulator).

### 3.2. Electrochemical Behaviour

Table 1 includes a summary of the cyclic voltammetry (CV) data collected in DCM (100 mV/s, 0.1 M NBu_4_PF_6_, 2 mM dye concentration), and a complete set of voltammograms (with and without ferrocene calibration) is included in the Appendix A. As mentioned previously, TPA donors show good electrochemical reversibility allowing us to elucidate the effect of appending various withdrawing or donating groups. While electrochemical reversible reductions were not observed, all the dyes in Groups 1–3 exhibit two well resolved electrochemical reversible oxidations, owing to the presence of two electronically unique redox-active TPA units. While it is a bit of an over-simplification, the oxidation wave at low potential is assigned to the distal TPA (because it is more electron-rich) and the second oxidation is centered on the proximal TPA unit (electron-deficient owing to the anchoring groups). This can be seen experimentally (Table 1) where distal substituents affected the first oxidation potential while the second oxidation potential exhibited a negligible change. Figure 6 elaborates this behavior observed in Table 1. Generally speaking, when considering the effect of substitution on the distal TPA, the donating ability of the substituent is consistent with what would be expected for traditional electrophilic aromatic substitution donors; such that -OMe ~ -OHex > -SMe > -H in terms of donating ability. Figure 6A highlights this trend visually. While only modest shifts in the second oxidation potential are seen; the strong donating effects of -OMe (**1b**) destabilize the HOMO compared to -H (**1a**) and -SMe (**1d**). 

The thiophene π-spacer has a more pronounced effect on the electronics of the dye that appears to manifest on the second oxidation potential and the resolution between the two TPA oxidation waves. Figure 6B demonstrates this behavior when considering the -SMe distal modification. The increased addition of thiophene spacers has a reciprocal mesomeric effect as it serves to be an electron rich donor to each TPA, but it also increases conjugation between the electron rich and electron poor TPA, more dramatically affecting the proximal TPA oxidation potential. This implies that additional thiophene π-spacers destabilize the HOMO − 1 to a greater extent, despite having two strong electron-withdrawing groups attached. Normally, thiophene π-spacers are thought to improve electronic communication between the two TPA units. However, if that over-simplification was the only effect at play, the second oxidation potential would increase. Instead the mesomeric effect creates a more delocalized HOMO-1 orbital that limits electrochemical interaction between donors. Regardless, this provides control of the oxidation potential of the proximal TPA. This is an incredible result, because it suggests that in this family, it is possible to tune the electronics of the two TPA units independently; spacers to tune the proximal TPA, and donating groups on the distal TPA unit. It is significant to note that a potentially deleterious behavior emerges with the addition of two π-spacers, as the HOMO energy of the dye increases (and consequently the rate of dye regeneration could decrease; vide infra). As a result, there is likely a sweet-spot for intramolecular charge transfer rates that may manifest as improved DSSC performance.

### 3.3. Photovoltaic Performance within the DSSC

The photovoltaic performance of the bipodal/bichromic family compared to benchmark dye **L1** is detailed in Table 2. DSSC cell fabrication, testing details and relevant current/potential (IV) curves have been included in the Appendix A. Device construction is done by hand (see Appendix A for fabrication methods) and multiple cells are made to get reliable statistics. The number of cells that make up the averages and standard deviations is shown as column ‘N’ in Table 2.

Pleasingly, in our hands, all of our bichromic-bipodal dyes outperformed benchmark **L1**, which further supports our previous study that our dye scaffold offers improved performance, with minimal increased synthetic effort. While, the DSSC dye interfaces represent a complicated collection of competing rates (recombination, injection, regeneration, etc.), when bichromic dyes are employed the rate of intramolecular electron transfer (Figure 7) adds an additional consideration. It is our theory that our improved performance is largely a result of rapid internal electron transfer, creating greater charge separation that facilitates dye regeneration before recombination [40]. While it is difficult to draw firm conclusions about dye performance, Markus theory predicts the existence of normal and inverted regions for electron transfer rates (related to difference in free energy changes). As such, each of these electron transfer rates could have a sweet spot for enhanced kinetics. While this adds complexity to the analysis, there are a number of important structure-property relationships that can be gleaned from this extended family. 

Table 3 summarizes the DSSC performance modulated by the nature of R substituents on the distal TPA unit and as a function of the number of thiophene spacers present. Group 1: **1a** ≈ **1d** > **1b** ≈ **1c**. Group 2: **2d** > **2a** > **2b** ≈ **2c**. Group 3: **3d** > **3a** > **3c** ≈ **3b**. From these trends, weak donors (-SMe & -H) stand out as superior distal substituents in this family of dyes. In addition, the effect of thiophene spacers on individual substitutes follows: -H: **1a** > **2a** > **3a**; -OMe: **2b** ≈ **1b** > **3b**; -OHex: **1c** ≈ **2c** > **3c**; -SMe: **2d** > **3d** > **1d**. This trend generally suggests that beyond 1 thiophene spacer, there is no significant increase warranting the added synthetic complexity. Below we examine these trends in greater detail. 

### 3.4. The Effect of the π-Spacer

When considering the DSSC performance data, while holding the distal TPA modification at parity, on aggregate, performance follows; 1 spacer ≥ no spacer > 2 spacers. Therefore, adding synthetic complexity to this dye family does not improve performance. As discussed previously, π-spacer extension largely influences the redox potential of the proximal TPA with minimal effects on the HOMO energy of the dye. When increasing the number of thiophene spacers, there is a decrease in the energy between first and second oxidation potential. This potential difference would likely have an adverse impact on the rate for internal energy transfer. Although a maximum would be expected from Markus theory, decreasing the energy difference between TPA units would lower the thermodynamic driving force for intramolecular e-transfer. By taking the difference between the first and second oxidation potential, it would appear that the potential difference sweet spot is ~300 mV in this dye family, a common number in e-transfer processes in the literature [58]. In addition, while it does not red-shift the absorption, the extinction coefficient for the dyes follows a similar trend; 1 spacer > no spacer > 2 spacers. Therefore, improved performance in this group could also be rationalized as having higher *J*_SC_ values from enhanced photon absorption. However, looking at those values (Table 2), that does not strongly corelate with this theory. Another possible argument is that owing to the mesomeric effects of the π-spacer, the increase in the HOMO energy decreases the dye regeneration rate with the electrolyte, resulting in the two thiophene spacer being the weakest performer. Yet another possible explanation for the poor performance of Group 3 is more pragmatic. These two thiophene spacer dyes are less soluble than the other families, and as discussed in the literature, planar dyes are more likely to aggregate intermolecularly which in turn lowers a DSSCs performance [54].

### 3.5. The Effect of Distal TPA Modification

When considering the DSSC performance data, the distal TPA modification follows -SMe > -H >> -OMe ~ -OHex. Considering the high number of literature dye motifs using -OR donors [18], this result should not be overlooked. While the -OHex poor performance could be rationalized as less dye loading or surfactant-like association, the same rationale cannot explain the performance difference between -SMe and -OMe derivatives (where -SMe is superior in all cases). The absorption data, shows no significant differences in the optical properties upon modifying the distal TPA, so likely the difference is routed in the dyes electrochemical behaviour. Consistent with the mesomeric influence described earlier, destabilizing the HOMO may have an adverse effect on dye regeneration rates, where moving the HOMO closer to the electrolyte redox couple, would decrease the thermodynamic driving force for regeneration.

While raising the dye HOMO energy too high could explain poor -OMe injection rates, it does not rationalize the superior -SMe performance over no substituent at all (-H). To understand this observation, we must further consider dye regeneration rates. It has been shown in literature that -SR donors have favourable “soft-soft” interactions with iodide-based electrolytes that enhance dye regeneration kinetics [59]. This behaviour typically manifests in having larger V_OC_ values. Looking at our V_OC_ data, this seems to be consistent, as -SMe derivatives in each family have the largest open circuit voltages. Not to be forgotten, the 3-thienyl-substituted TPA (**4**) did not have electrochemical or optical properties as desirable (high ε in the visible portion of the spectrum) as **1a** (-H) yet it still was one of the top performers. This can be rationalized consistent with our previous work [51] as high energy absorption behaviour (between 350–400 nm) is significant in organic dye performance (evidenced by *J*_SC_), and like -SR derivatives they are also shown to have favourable electrolyte interactions [53]. 

## 4. Summary and Conclusions

A family of bichromic/bipodal organic dyes with D-π-D-A motifs have been shown to improve DSSC performance efficiency relative to benchmark **L1** and a simplified summary of photovoltaic performance is presented in Table 3. While the influence of the π-spacer was seen electrochemically on the proximal TPA donor, its inclusion beyond 1 thiophene unit is not beneficial in this dye motif. The presence of various R substituents at the distal TPA donor region had a more pronounced role in changing the DSSC performance. The best performing DSSCs had dyes that contained -SMe, 3-thienyl, and -H substituents, respectively. This result was perhaps most surprising considering the ubiquitous nature of -OR substituents in the literature. These substituents allowed for the independent tuning of the distal TPA donor that ultimately improved regeneration and injection rates in this unique family of dyes. The DSSC performance of **4** was among the best due to a combination of high energy absorption behaviour and favorable regeneration rates. It is likely that combining this modification with one π-spacer could see an additional improvement owing to enhanced electronic communication between donors, but may lead to adverse solubility and dye aggregation. Considering the bio-inspired nature of these dyes, future work will focus on diffuse light absorption behaviour [15] and polymerizable hybrid device fabrication (using dye **4**) [10].

## Figures and Tables

**Figure 1 molecules-25-02260-f001:**
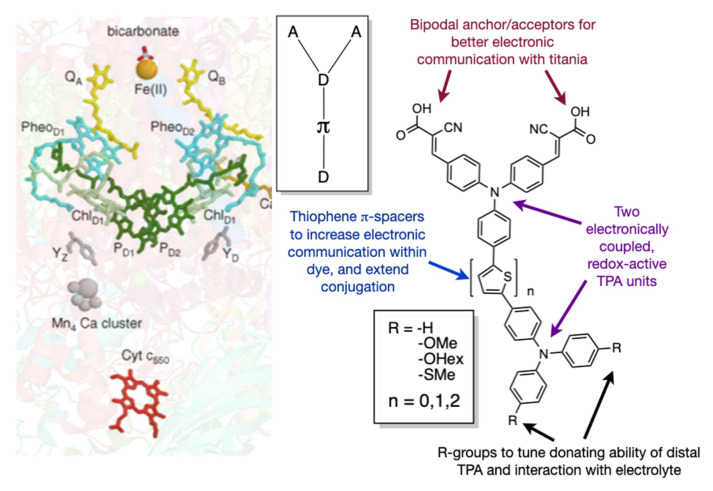
A family of bichromic-bipodal dyes inspired by the molecular machinery found in Photosystem II (PS2-left). Bichromic (two redox coupled donors) and bipodal (two acceptor anchors) (right) mimic the cofacial porphyrin dimer P680 and the two quinone acceptor pathways found, respectively.

**Figure 2 molecules-25-02260-f002:**
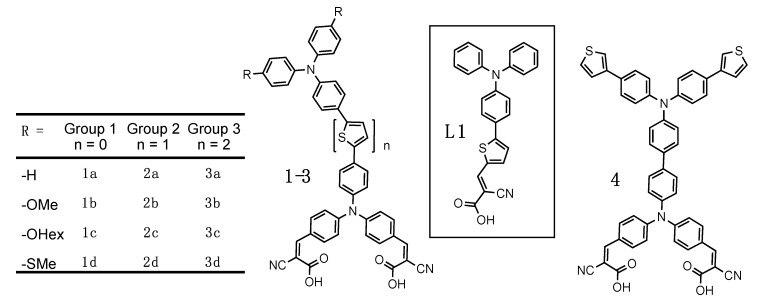
A family of bichromic-bipodal dyes; **L1** as benchmark; and **4** a dye with thiophene polymerizable units.

**Figure 3 molecules-25-02260-f003:**
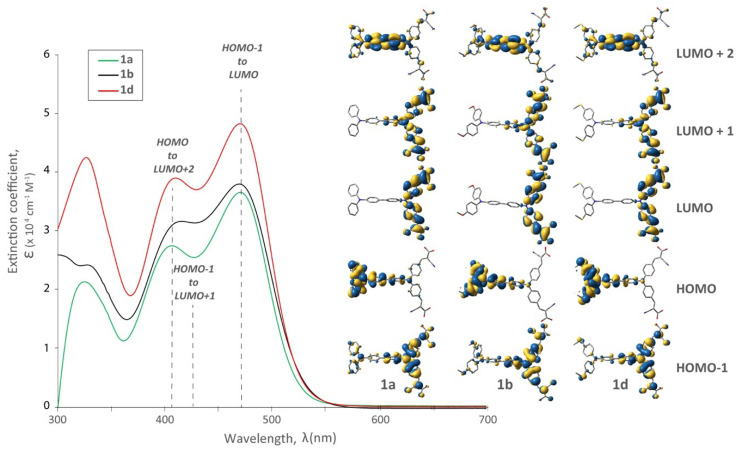
UV-Vis spectra and relevant molecular orbitals of bichromic/bipodal dyes (**1a**, **1b**, **1d**) with varied R substituents on the distal TPA while holding the remainder of the structure at parity.

**Figure 4 molecules-25-02260-f004:**
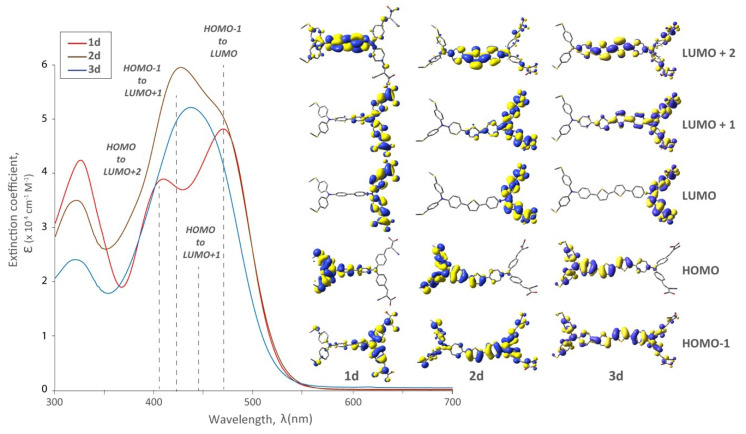
UV-Vis spectra and relevant molecular orbitals of bichromic/bipodal dyes (**1d**, **2d**, **3d**). The effect of extending the π-spacer on the distal TPA while holding the distal substitution constant with -SMe. This distal modification was chosen judiciously because these dyes produced the highest performing DSSCs (*vide infra*).

**Figure 5 molecules-25-02260-f005:**
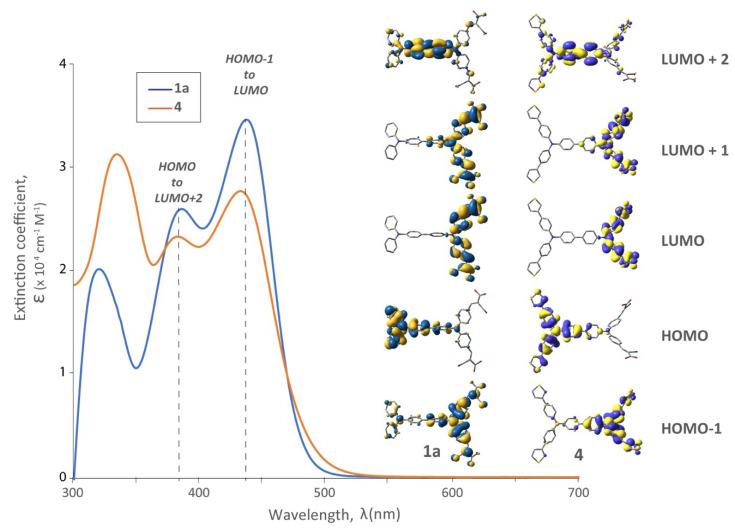
UV-Vis spectra of dye **1a** vs. **4** (3-thienyl substituents on distal TPA unit), and their respective DFT frontier orbitals.

**Figure 6 molecules-25-02260-f006:**
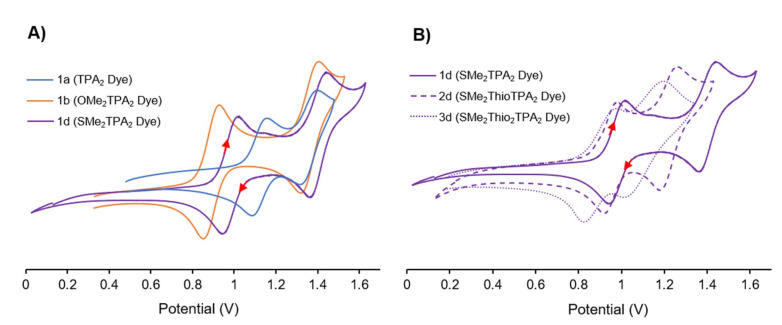
Examining substituent (**A**) and π-spacer effects (**B**) with cyclic voltammetry (DCM, 100 mV/s, 0.1 M NBu_4_PF_6_). A) In **1b** the strong donating effects of -OMe destabilize the HOMO of the distal TPA compared to **1a** or **1d**, while only modestly affecting the second oxidation potential. B) Contrasts the inclusion of additional π-spacers **1d** (none), **2d** (one), **3d** (two). While only modest changes are observed in the distal TPA oxidation potential, the inclusion of additional thiophene π-spacers more dramatically decreases the second oxidation potential.

**Figure 7 molecules-25-02260-f007:**
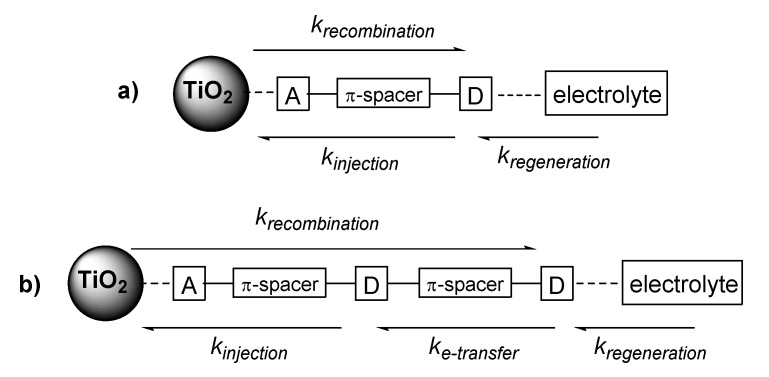
A simplified schematic of (**a**) traditional D-π-A dye motifs and (**b**) proposed bichromic dye kinetics.

**Table 1 molecules-25-02260-t001:** Physiochemical properties of DSSC dyes reported herein.

Dye	UV-Vis λ_max_ nm (ε × 10^4^ M^−1^cm^−1^)		E1/2 ^a^ (V vs. NHE)	
**L1**	483 (3.2) ^b^		1.23	
**1a**	471 (3.6)	407 (2.7)	1.12	1.36
**1b**	470 (3.8)	416 (3.2)	0.89	1.36
**1c**	423 (4.9)	302 (2.8)	0.87	1.34
**1d**	471 (4.8)	410 (3.9), 328 (4.2)	0.98	1.39
**2a**	421 (3.3)		1.04	1.19
**2b**	433 (6.2)		0.86	1.19
**2c**	415 (3.4)		0.84	1.14
**2d**	428 (6.0)	322 (3.5)	0.95	1.22
**3a**	436 (1.5)		0.98	1.14
**3b**	434 (0.7)		0.86	1.10
**3c**	437 (3.1)		0.84	1.10
**3d**	438 (5.2)	326 (2.4)	0.93	1.13
**4**	344 (3.3)	469(2.9), 399 (2.4)	1.07	1.43

^a^ Data collected using 0.1 M NBu_4_PF_6_ in DCM solutions at 100 mVs^−1^ and referenced to an octamethylferrocene [OFc]/[OFc]^+^ internal standard. Calibrated at 0.225 V vs. NHE for OFc [57]. ^b^ Values from previous work [52].

**Table 2 molecules-25-02260-t002:** Photovoltaic performance of DSSCs based on the extended bichromic-bipodal dyes, compared to benchmark **L1**.

Dye	V_OC_ (V)	*J*_SC_ (mA/cm^2^)	FF	η (%)	N
**L1**	0.51 ± 0.02	3.37 ± 0.32	0.58 ± 0.04	1.08 ± 0.13	7
**1a**	0.65 ± 0.02	5.58 ± 0.42	0.71 ± 0.02	2.77 ± 0.26	8
**1b**	0.64 ± 0.01	3.20 ± 0.64	0.70 ± 0.03	2.00 ± 0.35	7
**1c**	0.61 ± 0.03	4.57 ± 0.69	0.64 ± 0.05	1.90 ± 0.41	8
**1d**	0.66 ± 0.03	5.67 ± 1.00	0.66 ± 0.09	2.66 ± 0.68	8
**2a**	0.61 ± 0.01	5.16 ± 0.53	0.68 ± 0.03	2.31 ± 0.34	8
**2b**	0.68 ± 0.01	3.30 ± 0.43	0.67 ± 0.05	2.20 ± 0.23	7
**2c**	0.58 ± 0.02	4.30 ± 0.19	0.70 ± 0.03	1.90 ± 0.10	6
**2d**	0.71 ± 0.01	5.71 ± 0.84	0.72 ± 0.06	3.18 ± 0.63	9
**3a**	0.59 ± 0.02	4.38 ± 0.49	0.69 ± 0.03	1.90 ± 0.30	8
**3b**	0.54 ± 0.02	3.21 ± 0.56	0.64 ± 0.05	1.20 ± 0.21	8
**3c**	0.55 ± 0.03	3.26 ± 0.57	0.65 ± 0.05	1.29 ± 0.30	8
**3d**	0.66 ± 0.02	5.96 ± 0.73	0.66 ± 0.05	2.88 ± 0.21	8
**4**	0.63 ± 0.01	6.48 ± 0.75	0.65 ± 0.04	2.86 ± 0.41	7

N is the number of simultaneous test cells prepared that makes up the mean and standard deviation. Z1137 I^−^/I_3_^−^ electrolyte employed [see Appendix A].

**Table 3 molecules-25-02260-t003:** Structure-property relationship of TPA-based dyes via their average power conversion efficiency (η%).

π-Spacer	-H(η%)	-OMe(η%)	-OHex(η%)	-SMe(η%)	Dye 4(η%)
No thiophene spacer	**1a**2.77	**1b**2.00	**1c**1.90	**1d**2.66	**4**2.86
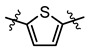	**2a**2.31	**2b**2.20	**2c**1.90	**2d**3.18	-
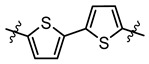	**3a**1.90	**3b**1.20	**3c**1.29	**3d**2.88	-

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
