# Peer review of "Exploring Structure-Property Relationships in a Bio-Inspired Family of Bipodal and Electronically-Coupled *Bis*triphenylamine Dyes for Dye-Sensitized Solar Cell Applications"

_molecules, 2020, doi:10.3390/molecules25092260_

Round 1

Reviewer 1 Report

The manuscript by Koivisto and co-workers faces up to a very interesting subject, from different point of views: the correlation between structure and properties of molecules. After having synthetized a considerable number of molecules (dyes and their di-aldehyde precursors), Authors characterized their optical properties (absorption and emission features) and electrochemical ones to try to depict an as much clear as possible portrait. Attention was mainly devoted to the effect of R substituents on distal triphenylamine units (TPA) and of the number of thiophene units (Th) as spacers between distal and proximal TPAs. Finally, such correlations plots were exploited to try to rationalize the photoelectrochemical performance of lab-scale DSSC employing a I-/I3- electrolyte.

The argument addressed by Authors deserves praise, for the intrinsic complexity of the theme (especially, as said by Authors, in the modelling of the processes occurring at the anode/electrolyte interface, determining the overall PCE of the devices). The work, in general, is acceptable and it is clear that Authors did a lot of works (from both synthetic, characterization and computational point of view). This is completely beyond doubtful. Unfortunately, in some cases, I had the impression that Authors didn't properly exploit the huge amount of the available data to depict a complete structure-property portrait of their molecules. In some cases (mainly in the electrochemical section), Authors statements are not properly supported by experimental data and some possible “cross-analyses” were not conducted, even if data allowed these valuable comparisons.

Authors must try to do this little step on to improve the paper; this is necessary before being considered for publication. Another mandatory improvement is related to the electrochemical section, probably the weak part of the manuscript. Moreover, improvements are necessary for the experimental details provided (or better, in some cases, not provided at all) and a bit more attention is requested in reporting the references cited. Here below I list my objections and my suggestions.

- My main criticism is related to the electrochemical section. Some lacks seems can be identified in the experimental work and in the interpretation of related data. Here below, I explain me better.

The main problem of this manuscript (and I’m sorry for this) is that electrochemical measurements were carried out in a not proper way, that could potentially have negatively affected all the related study. The “error” is in the procedure used to record CV patter of the analyte molecules (dyes and dialdehydes). Adding an “exogenous” molecule (ferrocene (Fc) or octamethylferrocene (Me8Fc), used as reference redox couples) in the working medium (especially if it reacts at lower potentials than the analyte itself) is not an acceptable way of working, simply because Fc o Me8Fc (or related oxidized form) can potentially interfere with the analyte, determining a CV patter not related to the real behaviour of the pristine analyte (in a blank solution). Due to these possible interferences, without a proof that no interactions can occur between analyte and reference redox couple, all electrochemical data reported here are not acceptable. I'm very sorry for this.

To prove the "innocent" role of Fc or Me8Fc, Authors must report, for each molecules analysed, a CV recorded in a Fc-free or Me8Fc-free solution, too. These two curves (with and without reference redox couple, for each analyte) must be reported in the same plot, overlapping them to estimate if any modification to the analyte curve was obtained or not (in term of peak position, for example).

Additional weak aspects of the electrochemical section are reported below:

* Authors state “reversible oxidation” (e.g., pag 7, line 172) many times. Actually, in electrochemistry two types of reversibility exist: electrochemical reversibility and chemical reversibility. It’s better to specify which is pertinent here.

*pag 7, line 176: “…simplification, the 176 oxidation wave at low potential is assigned to the distal TPA and the second oxidation is centered on 177 the proximal TPA unit”. I agree with Authors, but Authors must clearly explain how they attributed the first redox process to distal TPA and the second one to the proximal TPA. Which is the experimental observations that allow them to state this? For non technical readers, I think this is not clear.

* pag 8, line 193: “…thought to improve electronic communication between the TPA units”. It is right. But thiophenes have also mesomeric effects, that stabilize additional charges too. Authors should properly consider this effect, too. Moreover, to definitely prove that the modulation of the second Ox process is due to the different electronic communication between the two TPA units, again electrochemistry provides a way to prove this. Try to add some acetonitrile or dimethyl formaldehyde, and see what happens.

* pag. 8, line 196-197: “This is likely a consequence of a greater spatial distance between the two redox-active units”. Again, this sentence must be properly supported. There is a way (through electrochemistry) to support if the spatial distance is crucial of not. See the above suggestion.

* pag 8, line 199-200: “the inclusion of a second thiophene π-spacer […] further amplifies this observation”. Why the effect of a second Th unit is not considered keeping constant the distal substituents? Table 1 shows that this is possible. So, please do this, adding in Fig 6c the CV curve of 3b. And properly adapt the discussion. In this way, a more rigorous discussion on the effect of the -Th units will result.

* Authors firstly discussed the effect of R substituents and of Th units, separately. Another important consideration that Authors must do is to rationalize the modulation of the electron releasing effect of R groups on the distal TPA unit induced by the number of -Th units. Authors did a lot of valuable works, so it’s better to exploit as much as possible them, to offer to readership a complete portrait derived by any possible intersection of the data.

* Figure 6: in Fig. 6a I suggest to add also the curve of 1 d, and in Fig 6b that of 2a. In this way, it is easier to judge, even by eye, the electron releasing effect of substituents (OMe, SMe, H) in both families. Moreover, in all CV plots, please increase the unit resolutions along the X axis (e.g., each 0.2 V). This made easier estimation of the potential shifts discussed in the text.

* Another “minor” observation, concerning electrochemistry: why Authors used two different redox couples for potential referring (Fc e octamethyl-Fc)? And more importantly: not expert readers can mislead the role of this redox couple. In footnotes of Table 1 you said: ”… referenced to a octamethylferrocene [OFc]/[OFc]+ internal standard for the dyes”. This sentence is not formally correct from an electrochemical point of view. The couple you referred is not a reference electrode but an intersolvental redox couple. So, Authors must report the actual operative reference electrode use while recording all the CV curves.

* Electrochemical impedance spectroscopy study was performed and the results are reported in SI file, together with equivalent circuit and fitting. Why Authors did not discuss the results in the main text? This is not acceptable.

Other weak points, encountered in the manuscript:

- Table 1 (and related sections in the main text): Actually, E0-0 is the 0-0 transition energy, so the energy difference between the ground state and the excited state of a molecule. This is not properly the HOMO-LUMO gap, as said by Authors. HOMO and LUMO are referred to levels of a molecule in its ground state. So please, amend this inconsistency. Moreover, looking at the spectra in Supp. Info file, due to the low signal/noise ratio of emission spectra, the absorption and emission one are not properly normalized (the maximum of the emission spectra are, sometimes, >1 or <1). Due to the poor emission properties of the molecules, maybe evaluation of E0-0 is too much approximated. I suggest to Authors to consider the possibility to delete this column, being too much approximated (or at least to clearly state that these values are only qualitative).

- Pag 5, from line 136 “The presence of thiophene π-spacers had a more pronounced effect…”. Why Authors decide to discuss the thiophene rings role using the d molecules (Fig 4), that are the unique of the series that present a "non-innocent" -SMe substituent (as they previously said)? I think it is definitely better to do the same discussion using one of the others series (a, b or c), for which no potential interferences from the peculiar -SMe group can be expected. I strongly suggest Authors to do this.

- Concerning Fig 7 (and discussion of the related thesis): the scheme is essentially correct, but it's probably oversimplified. Actually, it does not take into account a very important aspect, that significantly weaken the here reported thesis. That is: recombination occurs also (and mainly) directly between TiO2 (or exposed underlying FTO) and the oxidized form of the redox mediator in solution. So, Authors must consider this aspect in the discussion.

- In section 3.4: discussing the effect of Th units on DSSC performance, an aspect was not properly taken into consideration by Authors, but it must be: the driving force for dye regeneration (as difference between the mediator and the dye redox potentials). According to Table 1, in some cases, by increasing the number of Th units, the E1/2 of the dye is negatively shifted up to 140 mV. This means a significant reduction in the driving force for dye regeneration (by I- species). Authors should also consider this aspect in the discussion, to better analyse DSSC data in Table 2.

- pag 11, line 270:” As a result, this may have an adverse effect on dye injection rates, because the LUMO would move far above the conduction band edge of TiO2”. Authors must provide some references testifying that a higher electron injection driving force results into worst injection phenomenon. To the best of my knowledge, the opposite is correct: lower the driving force for injection, worst is the e- injection into TiO2 from excited state of the dye.

- To improve the validity of the discussion on DSSC performance (in particular, in 3.5 section): according to my previous suggestion for the electrochemical part, Authors should discuss how the effect on DSSC performance is modulated by the nature of R groups on distal TPA unit as a function of  the number of Th units used as spacer (I mean, in Group 1: 1a>1d>1b>1c; in Group 2: 2d> 2a> 2b> 2c; Group 3: 3d> 3a> 3c> 3b).

- Other important weak point: the manuscript completely lacks any “Experimental Details” section. Authors reported a “2. Synthesis” section, that actually does not report any synthetic procedures, or something related to this. An “Experimental Details” section is mandatory in any scientific paper. So, first of all, please move the text in "General Consideration" section in the Supp. Info into the main text. But, this is not enough. Some aspects discussed in the manuscript lack completely of the related experimental detail section. The computational work is an example: no details are provided at all. Moreover, for electrochemistry section (in Supp. Info) too poor details are provided: which is the solvent? Which is the working electrode? The counter one? Which is the potential scan rate used?

All these aspects must be fixed before the resubmission.

Minor observations:

* About references: the first one is not referred to perovskite cells, as it should be according to the text cited. Similar consideration for ref. 8: it refers to porphyrin dyes that are not properly "organic dyes". Please amend.

* pag 2, line 56: “…, we present a modified dye topology with a D-π-D-A motif that has greater control on the redox potential of the distal TPA”. Actually, the topology is identical to that proposed by Authors themselves in ref. 19. Authors simply add an additional thiophene unit (Group 3) and changed R groups on the ending TPA unit. By the way, the "structure" is common to dyes in ref 19. So, please better clarify this sentence.

* pag 4, line 97: Figure S2 is cited before Figure S1 (line 105). This is not acceptable. Please, fix it.

* pag. 4: looking at Table 1, why -OHex chains invariably determine a decrease in the wavelength of maximum absorption in Group 1 and 2? Why does not the same happen for Group 3?

* pag. 6, line 144: “…which results in a coalescence between the donors and the dye absorption profile”. I found the sentence not clear: what do you mean with “donor absorption profile” and “dye absorption profile”? The donor is part of the dye; how do you define "dye absorption"?

Reviewer 2 Report

The manuscript by Al-Faouri et al. nicely investigate a set of bio-inspired bistriphenylamine dyes for DSSC. The implementation of these systems into DSSC is also reported and discussed. Though the manuscript reprents an extension of a previous paper by the same  authors (as stated by the authors themselves), which somehow limits the novelty of the investigation,  this topic is of interest to the reader because it allows to gain insights into the molecular structure-electronic properties-device response relationships, which is needed to design ever more efficient systems. 

A few issues, which are listed below, must be however addressed before publication in Molecules Journal:

-More updated references should be included in the Introductory part, to better highlight both the fact that reaserch in photovoltaics has shifted towards perovskite and that DSSC are still of interest to the scientific community

-Reporting the range of the highest efficiencies of dssc to date (and associated references) would also be beneficial

Reviewer 3 Report

The authors present a new family of dyes with a D-π-D-A structure to investigate the effects of different substituents in the outer TPA unit and of different π bridge lengths on the properties of said dyes and on the performance of DSSCs sensitised with them. The design of this work is very well-thought and the experimental part well executed, with extensive characterisation carried out to elucidate and rationalise the properties and behaviour of all investigated dyes.
However, before this work can be considered for publication, there are minor and major concerns that should be addressed, listed as follows:

  • In section 3.4 the authors state that DSSC efficiency follow the trend 1 spacer > no spacer > 2 spacers. However, looking at the data in Table 2 and Table 3, this is only true for the d and b substituents, while it's not really true for the c substituent and it's definitely very wrong for the a substituent. Therefore, the analysis and subsequent conclusion for this part is wrong (or at the very least too general/simplified) and should be revised.
  • Regarding the UV-Vis spectra in Figures 3, 4, and 5, the absorption peak at lowest energy is usually associated to HOMO/LUMO transitions, as they are those with the narrowest gap, while the authors associate them to either to HOMO-1/LUMO or HOMO/LUMO+1 transitions. If the authors think that these different transitions are actually the correct ones, they should provide an explanation for them in the text. Furthermore, for said peak of compound 1d a HOMO-1/LUMO transition is proposed in Figure 3, while a HOMO/LUMO+1 transition is proposed in Figure 4, which is inconsistent.
  • About cyclic voltammetry data in Table 1 and CV plots throughout the main text and supporting information, I understand that the shape of the voltammogram and the peak potentials are all that is needed for the discussion of relevant dye properties. However, I know from personal past experience that a proper electrochemist would look at this data with great distaste, because key information is withheld that would enable reproduction of this data. Therefore, if this information is available I suggest that you specify dye concentration in solution during CV experiments, and I definitely recommend adding the y axis to all voltammograms, to give an indication of the current values.
  • In the caption of Figure 4 the words "R substituents" were kept during the copy/paste from the previous figure's caption, but they should be removed.
  • The aim of Figure 6 should be to give a visual representation of the effects of the different substituents or π bridge lengths on the redox properties of the dyes, but the visualisation is confusing and unhelpful. The authors already provide a comprehensive plotting of such data in the SI, and there is no reason do draw as many voltammograms as possible in the main text. To give the reader a better visual representation, I suggest to limit Figure 6 to two plots, one for the substituents and one for the thiophene units. In the first one there should be all three voltammograms for the different substituent in the same core dye structure (e.g. 1a, 1b, 1d or 2a, 2b, 2d or any other trio) and in the second there should be all three voltammograms for the different number of thiophene units, with the same substituent (e.g. 1b, 2b, 3b or 1d, 2d, 3d or any other trio).
  • Within the SI, in the description of the synthesis methods, "Pd catalyst", "phosphine ligand" and "phosphonium ligand" should be replaced with the actual names of the chemicals used for the synthesis, to facilitate identification of said chemicals by the readers.
  • The whole section in SI regarding EIS is never mentioned in the main article. I am in no way an expert in impedance, but the plots and associated data for each dye look different enough to suggest that they can provide some insights in the differences between each dye. If this is the case, this should be discussed in the main article. If instead they show that all dyes are quite similar, this should also be mentioned. If the authors think that it is not worth discussing EIS results in the main article, then they should probably remove that data from the SI as well.
  • Although the article is overall well written, there are several minor or not so small English mistakes throughout the text, together with some typographical issues (such as double spaces). I suggest a careful reading of the current text to fix all grammar issues.
  • I suggest that you rework some of your plots. At the very least, try to not have the x axis drawn in the middle of your plotted curve (e.g. in the voltammograms)

Round 2

Reviewer 3 Report

I thank you the authors for their revision work, the text reads much better now, and the figures are better presented.

However, I still have to argue about HOMO/LUMO levels, and the inconsistency in figure 3/4 that you haven't addressed.

About the whole orbital/UV-Vis peak assignment, I accept your explaination for it, but I would like to see it clearly stated and explained in your manuscript, rather that just writing it in the reply to the reviewer. Your peak assignment was unclear to me when reading the article the first time, and it's still unclear to me in the current version, if I disregard the information contained in your reply to me. Can you please make the text a bit more clear in the manuscript?

About the inconsistency, judging from your answer, I think I haven't explained myself well enough, so I'll try to be more clear now, with screenshots of your figures (if the MDPI system allows me). This is a portion of Figure 3:

and you can see that the rightmost peak of compound 1d is assigned to a HOMO-1/LUMO transition.

This is a portion of figure 4:

and you can see that now the same peak is assigned to a HOMO/LUMO+1 transition.

Now, I understand that "The transitions do swap in the p-spacer system owing to the presence of the thiophene units", but the same peak for the same molecule should always originate from the same transition, not from different ones depending on the figure. I hope you can agree with this?

The English has been greatly improved, and there are now only a couple of minor issues that can be fixed at editorial level. However, I've noticed that here and there, in your sentence reworking, you have deleted a few too many words, making some sentences a bit weird. When you're going to accept revisions in Word at the end of the reviewing process, make sure that you accept them one by one rather than all together, and have a look at what you're exactly inserting/deleting. I'm pretty sure the extra deletions will become evident then.

Author Response

Thank you for the clarification. Now I understand what you were referring to between figure 3 and Figure 4. While I did misunderstand your initial comment/clarification, I am a little embarrassed that I did not notice that red flag earlier. After re-examining the TDDFT calculations, it appears there was a typo in Figure 4, and the peaks were mis-labelled in the graphic (assignment labels were reversed). We have triple checked the other assignments and  have clarified the text to address this discrepancy, and your other comment about adding more detail to this section. Thank you again,  for taking the time to re-clarify so we could address that error!